# Advances in Liposomal Drug Delivery: Multidirectional Perspectives on Overcoming Biological Barriers

**DOI:** 10.3390/pharmaceutics17070885

**Published:** 2025-07-05

**Authors:** Żaneta Sobol, Rafał Chiczewski, Dorota Wątróbska-Świetlikowska

**Affiliations:** Department of Pharmaceutical Technology, Pomeranian Medical University in Szczecin, Rybacka 1, 70-204 Szczecin, Poland; zaneta.sobol2000@gmail.com (Ż.S.); r.chiczewski@o2.pl (R.C.)

**Keywords:** liposomes modification, targeted therapy, nanocarrier-based drug delivery systems

## Abstract

Liposomes represent a cornerstone of modern drug delivery systems due to their unique structural and physicochemical characteristics. Extensive research has refined their formulation, stability, and targeting capabilities, leading to numerous clinical applications, particularly in oncology. A key clinical feature is their ability to accumulate in malignant tissues via the enhanced permeability and retention effect, offering improved pharmacokinetics and reduced systemic toxicity. Advances in liposomal engineering, including PEGylation and ligand-based targeting, have significantly enhanced pharmacokinetic profiles and tissue specificity, minimizing off-target toxicity. The modern approach to nanocarrier-based drugs offers multidirectional perspectives on targeted therapy. Liposomes can bypass drug resistance mechanisms and provide controlled or stimuli-responsive drug release. Current trends in liposome research focus on hybrid nanocarriers, personalized medicine applications, and combination therapies.

## 1. Introduction

Nearly 90% of newly discovered active pharmaceutical ingredients (API) are classified as poorly soluble and belong to Class II of the Biopharmaceutics Classification System (BCS) [1]. The bioavailability of these drugs is directly linked to their dissolution rate. Improvement in API solubility is a crucial step toward enhancing therapeutic efficacy, which has driven the development of nanoscale drug delivery systems. Among these, liposomes have emerged as a cornerstone technology due to their versatility, biomimetic architecture, structural similarity to cellular membranes, and high biocompatibility. Liposomes are spherical nanocarriers composed of one or more concentric lipid bilayers enclosing an aqueous core. Their amphiphilic nature allows them to encapsulate a wide variety of therapeutic agents, including small hydrophobic drugs, hydrophilic macromolecules, nucleic acids, and peptides. This unique capability positions liposomes as ideal vehicles for improving drug solubility, stability, and site-specific delivery. These advantages are particularly relevant when considering the challenges of drug delivery across biological barriers. Biological barriers for drugs include the gastrointestinal tract, which limits absorption due to acidic pH, digestive enzymes, and the protective mucus layer. The blood–brain barrier (BBB) presents another major challenge, as it tightly regulates the passage of substances into the central nervous system through tight junctions and efflux transporters. Additionally, the reticuloendothelial system (RES) and immune responses can rapidly clear or inactivate drugs before they reach their target tissues. The gastrointestinal environment presents risks of enzymatic degradation, while systemic circulation subjects drug carriers to clearance by the reticuloendothelial system. Biological barriers such as the tumor microenvironment and the blood–brain barrier further complicate effective drug delivery. The enhanced permeability and retention (EPR) effect, often present in tumors and inflamed tissues, facilitates the passive accumulation of nanocarriers such as liposomes. Conversely, the BBB is a highly selective barrier that restricts the entry of most therapeutic compounds into the central nervous system. To overcome these obstacles, modern liposomes are increasingly being engineered with responsive features—such as stealth properties, ligand-mediated targeting, and stimuli-sensitive release mechanisms—allowing them to reach previously inaccessible pathological sites [2]. Nanoscale drug delivery systems have emerged as transformative tools, significantly altering the pharmacokinetics of drugs, their therapeutic effects, and safety profiles. Nanocarrier-based drug delivery systems require rigorous physicochemical assays due to their complexity. According to international regulatory standards, key parameters such as the particle size, zeta potential, and solubility must be assessed, as they serve as critical indicators of both efficacy and potential toxicity. The *Noyes–Whitney* equation provides insight into why nanocarrier-based drug delivery systems are so effective [3]. A reduction in particle size leads to an increased specific surface area, which proportionally enhances dissolution rates, thereby improving the absorption of poorly soluble drugs (Figure 1). Active pharmaceutical ingredients can be incorporated into nanoscale delivery systems, including liposomes. These advanced formulations not only protect drugs from degradation within the gastrointestinal tract but also ensure their targeted and efficient delivery to specific sites within the body. Liposomes tend to accumulate passively in pathological tissues by leveraging the enhanced permeability and retention effect. In tumors, as well as in sites of infection and inflammation, vascular structures typically become more permeable and exhibit wider endothelial gaps than those found in healthy tissue. Nanocarriers such as liposomes, typically ranging from 60 to 150 nm in diameter, can extravasate through these permeable blood vessels into diseased tissue, where they localize and gradually release their drug cargo. Due to deficient lymphatic drainage within tumor tissues, liposomes are retained more efficiently, leading to higher local concentrations of the therapeutic agent [4]. This property offers a distinct advantage over conventional, non-encapsulated drugs, which often distribute indiscriminately to both healthy and affected tissues. Upon reaching the pathological site, the encapsulated drug passively diffuses through the liposomal bilayer, disperses in the interstitial matrix, and is subsequently taken up by target cells [4,5,6].

A major obstacle in the development of central nervous system therapies is that most of the particles are unable to cross the blood–brain barrier [7]. The blood–brain barrier is designed to limit the passage of xenobiotics through its highly selective endothelial layer, tight junctions, enzymatic defenses, and active efflux transporters (Figure 2). One of the most significant advantages of nanoscale drug delivery systems is their ability to cross biological barriers. Nanocarriers, including liposomes, offer a promising solution by facilitating drug transport across this barrier via mechanisms such as tight junction modulation, receptor-mediated transcytosis, and endocytosis. The tight junctions formed by brain microvascular endothelial cells regulate paracellular transport, whereas transcellular transport is regulated by specialized transporters, pumps, and receptors. Nanocarriers can open tight junctions between endothelial cells, transcytosis through the endothelial cell layer, and endocytosis by endothelial cells, releasing the drug inside the cell [8]. These properties offer new opportunities for treating neurological disorders, which have been hindered by the inability of conventional drugs to effectively reach their targets within the brain. Recent advancements in pharmaceutical research have increasingly emphasized nanostructures, which offer huge potential in targeted therapy. The development of nanotechnology allows for modifications to drug delivery systems that revolutionize current therapies by increasing the bioavailability, solubility, and targeted transport of active substances to the site of action. Liposomes are spherical nanostructures consisting of lipid bilayers that enclose the aqueous phase inside them. The liposome shell is constructed analogously to biological membranes. Due to their structure, they can effectively transport both hydrophilic and hydrophobic active substances. Liposomes demonstrate high biocompatibility and the ability to fuse with biological membranes, which enables the effective release of drugs directly to the site of action and limits their exposure in other parts of the body. Liposomes can be categorized based on their structure, size, and technique of preparation. Structurally, they are divided into unilamellar, oligolamellar, and multilamellar vesicles, depending on the number of lipid bilayers. Unilamellar vesicles consist of a single lipid bilayer, typically measuring between 50 and 250 nm in diameter. Due to their prominent aqueous core, they are well-suited for encapsulating hydrophilic drugs. Multilamellar vesicles feature multiple concentric lipid layers with sizes spanning from 1 to 5 μm. Their high lipid content makes them particularly effective in entrapping lipophilic drugs through passive encapsulation. Due to their advantages, liposomes are the starting point of many scientific studies and constitute the basis for the modification of modern nanocarriers [9]. Scientific publications include liposome modifications such as stealth liposomes, pH-sensitive liposomes, thermosensitive liposomes, thermosensitive magneto-liposomes, and enzyme-responsive liposomes.

## 2. Enhancing Circulation Time—Stealth Liposomes

A major breakthrough in the development of liposomes with prolonged circulation in the bloodstream was achieved with the integration of polyethylene glycol (PEG) into the liposome structure [7,8]. The presence of PEG significantly extends the circulation time in the bloodstream while simultaneously reducing uptake by the mononuclear phagocyte system. This advancement improved both target specificity and therapeutic efficacy. Modifications of PEG terminal groups allow for the attachment of monoclonal antibodies or ligands, enabling active targeting of specific cells (Figure 3). PEG is biocompatible, water- and organic solvent-soluble, and it has low toxicity, minimal immunogenicity, and favorable excretion kinetics. Among the various polymers explored to prolong the liposome circulation time, PEG has emerged as the most widely used steric stabilizer due to its unique physicochemical and biological properties.

PEG can be incorporated into liposomes using multiple techniques [10]. The most common method involves anchoring the polymer within the lipid bilayer using a cross-linked lipid. As a biocompatible polymer, PEG enhances drug solubility, improves pharmacokinetics, and reduces immunogenicity. Nowadays, widespread use of PEGylated drugs is observed in modified therapeutic proteins and peptides, such as PEG-adamase (for immunodeficiency), PEG-asparaginase (for leukemia), PEG-interferon-alpha (for hepatitis C), and PEG-filgrastim (for neutropenia in chemotherapy patients). The attachment of PEG to liposomes can be achieved through various strategies, including physical adsorption onto the vesicle surface, the incorporation of PEG-lipid conjugates during liposome synthesis, or covalent attachment to preformed liposomes. The behavior of PEGylated liposomes depends on factors such as molecular weight, the surface density of PEG chains, and the overall polymer conformation. All properties influence circulation longevity and interactions with biological environments. These parameters determine surface coverage and spacing between PEG chains, affecting the overall stability and biodistribution of the liposomes. The most notable feature of PEGylated liposomes is their prolonged circulation time, which remains consistent regardless of surface charge or the inclusion of stabilizing agents such as cholesterol [11,12].

While PEGylation confers numerous pharmacokinetic benefits—including extended systemic circulation and reduced recognition by phagocytic cells—recent studies have identified significant limitations. These include the so-called “Accelerated Blood Clearance” (ABC) phenomenon, whereby the repeated administration of PEGylated nanocarriers induces anti-PEG IgM production and subsequent rapid clearance. Additionally, PEG chains may create steric hindrance that impedes receptor–ligand interactions on the liposome surface, thereby reducing the efficiency of active targeting strategies. To mitigate these effects, researchers have investigated a wide array of PEG alternatives. Notable examples include poly (zwitterions), which resist nonspecific protein adsorption; poly (2-oxazoline)s (POx), offering stealth properties with reduced immunogenicity; and polyglycerols, known for their tunable hydrophilicity and high chemical stability. Natural polysaccharides such as hyaluronic acid and dextran are also being employed due to their innate biocompatibility and receptor-mediated targeting capabilities. In addition, advancements in ligand-functionalized PEGylated liposomes have led to the development of highly specialized drug carriers. Conjugation strategies include the following:Tumor-targeting peptides, such as RGD motifs, that bind to integrin receptors.Cell-penetrating peptides (CPPs) like TAT or penetratin that facilitate endosomal escape.Antibody fragments such as Fab or scFv directed against tumor-associated antigens (e.g., HER2 or EGFR).Protein–lipid fusion constructs that respond to tumor-specific enzymatic activity.

These innovative modifications broaden the therapeutic scope of stealth liposomes and improve their ability to accumulate at and penetrate target tissues.

One of the most significant applications of PEGylated liposomes is oncology treatment, where passive targeting mechanisms are exploited to accumulate preferentially in tumor tissues—a process known as the enhanced permeability and retention effect [9,10]. Despite the ability to reach tumor sites, stealth liposomes exhibit a limited ability to penetrate cancer cells; instead, they remain in the extracellular tumor environment. Drug release from the liposome into the tumor interstitial fluid is essential to ensure therapeutic efficacy. Factors such as liposome stability, membrane composition, and lipid fluidity influence drug release dynamics. PEGylated liposomal doxorubicin, marketed as DOXIL^®^/Caelyx^®^, was the first stealth liposome formulation approved in both the USA and Europe for Kaposi’s sarcoma and recurrent ovarian cancer [13]. Currently, DOXIL^®^ is being evaluated in clinical trials for other malignancies, including multiple myeloma, breast cancer, and gliomas, often in combination with chemotherapy agents such as paclitaxel, docetaxel, temozolomide, and vinorelbine.

## 3. Targeted Drug Delivery—pH-Sensitive Liposomes

pH-sensitive liposomes are specialized to release their therapeutic cargo in response to pH fluctuations in the surrounding environment. These liposomes leverage changes in acidity to trigger drug release, making them particularly effective in delivering pharmaceuticals or genetic material into cells through the endocytotic pathway [14]. pH change allows for precise targeting of pathological tissues such as tumors, metastatic sites, inflamed regions, ischemic areas, and infection sites, where the pH is lower than normal physiological levels [15,16].

pH-sensitive liposomes can be formulated by either introducing pH-responsive units into preformed liposomal dispersions or by incorporating pH-sensitive lipids and polymers during vesicle assembly. While remaining stable at physiological pH, they undergo structural destabilization and become fusogenic in acidic microenvironments (approx. pH 5.3) (Figure 4). The controlled release of encapsulated drugs directly into the cytoplasm significantly improves therapeutic targeting while protecting the cargo from degradation within lysosomes [14,15]. Key components for designing pH-sensitive liposomes include polymorphic lipids such as unsaturated phosphatidylethanolamine (PE) and its derivatives, including diacetylenic-phosphatidylethanolamine (DAPE), palmitoyl-oleoyl phosphatidylethanolamine (POPE), and dioleoyl-phosphatidylethanolamine (DOPE). To enhance stability at a neutral pH, DOPE is often combined with mildly acidic amphiphiles, such as oleic acid (OA), cholesteryl hemisuccinate (CHEMS), and palmitoyl homocysteine (PHC). These molecules undergo protonation in acidic conditions (such as inside endosomes or lysosomes), reducing their hydrophilic volume and leading to membrane destabilization, which in turn facilitates the release of encapsulated bioactive compounds.

Studies evaluating pH-sensitive liposomes have demonstrated that they are capable of releasing 60–90% of their encapsulated drug cargo within 2 to 6 h in environments with pH values ranging from 5.0 to 5.5—conditions mimicking early endosomes or acidic tumor niches [17,18]. The release kinetics are highly dependent on the lipid composition, PEGylation density, and the physicochemical properties of the encapsulated agent. Notably, the use of unsaturated phosphatidylethanolamine derivatives in combination with pH-labile amphiphiles has been shown to enhance the destabilization of the bilayer structure, thereby promoting efficient cytoplasmic delivery [17].

Some pH-sensitive liposomes contain N-acyl-phosphatidylethanolamine derivatives, which additionally enhance membrane fusion. Other compositions, such as N-citraconyl-dioleoyl-phosphatidylethanolamine (C-DOPE) and N-citraconyl-dioleoyl-phosphatidylserine (C-DOPS), exhibit reversible structural transitions that increase permeability. This is achieved through covalent modifications of nucleophilic functional groups on lipid headgroups or the cleavage of alkyl chains, exposing long fatty acid chains that disrupt membrane stability and facilitate drug permeation. A promising advancement is the use of a poly-(ethyleneglycol)-N-distearoylphosphatidylethanolamine (PEG-DSPE) conjugate with a pH-sensitive linkage. The inclusion of PEG on liposome surfaces minimizes uptake by the reticuloendothelial system (RES) and prolongs the circulation time in the bloodstream. Under a neutral pH, proteins or peptides carried by these liposomes remain inactive, but upon encountering an acidic environment, conformational changes in these biomolecules promote fusion between the liposomal and cellular membranes. This pH-triggered membrane destabilization, facilitated by either full proteins or peptides, enhances the efficiency of liposomal drug delivery and lipid-based transport systems, making them highly adaptable for targeted therapeutic applications [14,18,19].

## 4. Targeted Drug Delivery—Thermosensitive Liposomes

Thermosensitive liposomes (TSL) are designed to enhance localized drug delivery when combined with mild hyperthermia (40–43 °C). This approach allows for effective targeting of tumor tissues while minimizing systemic toxicity. When reaching a heated tumor site, they undergo structural changes that rapidly release their encapsulated drug load (Figure 5 and Figure 6). This approach has been widely explored in cancer treatment and is also being investigated for other applications, including antibiotic delivery, treatment of inflammatory diseases, and blood clot resolution [20,21]. TSLs can release their drug cargo via two distinct mechanisms: extravascular triggered release and intravascular triggered release. Extravascular release relies on the enhanced permeability and retention (EPR) effect, where TSLs accumulate in the tumor interstitium and subsequently release their drug upon exposure to hyperthermia. However, the EPR effect has limitations, including high variability between tumors and an upper limit on drug accumulation. In contrast, intravascularly triggered release occurs within the microvasculature when TSLs pass through the heated tumor region, eliminating the need for EPR-dependent accumulation. This strategy has demonstrated superior drug delivery efficiency, with up to 25 times higher drug concentrations in tumors compared to unencapsulated drugs. Recent studies suggest that intravascular release achieves significantly higher tumor drug uptake than non-triggered nanoparticle drug delivery systems and is more effective than extravascular release approaches [20,21].

For thermosensitive formulations, rapid and complete drug release upon reaching the target temperature is critical to therapeutic efficacy. Lysolipid-containing TSLs (LTSLs) have demonstrated the ability to release over 95% of their drug load within 2–3 s when exposed to mild hyperthermia (approx. 42 °C). However, a major drawback of first-generation TSLs is their tendency to exhibit partial leakage at physiological temperatures (approx. 37 °C), which compromises systemic stability. More recently, the incorporation of novel lipid components such as DPPG2 has enabled the development of second-generation TSLs with improved circulation times and more controlled heat-triggered release. These advanced formulations have shown plasma half-lives of 5–10 h in preclinical models while maintaining rapid release profiles under localized heating.

Early tested TSL formulations, composed of dipalmitoyl phosphatidylcholine (DPPC) and distearoyl phosphatidylcholine (DSPC), were developed based on their low transition temperatures (approx. 41 °C). While these lipids facilitated temperature-dependent drug release, early TSL formulations, later termed traditional thermosensitive liposomes (TTSL), exhibited several drawbacks:Rapid Clearance—TTSLs were eliminated from circulation within an hour due to uptake by the reticuloendothelial system (RES).Slow Drug Release—The release rate was insufficient for efficient intravascular drug delivery.High Activation Temperature—Drug release only occurred at 43–45 °C, which risks tumor perfusion shutdown.Leakiness at Body Temperature—TTSLs were unstable at 37 °C, leading to premature drug loss before reaching the target site.

To improve stability and the circulation time, PEGylation and liposome size optimization were introduced, reducing RES clearance. The addition of cholesterol also enhanced stability but delayed drug release. For intravascular triggered release, rapid drug release is essential, as TSL only remains in the heated tumor microvasculature for seconds. To overcome limitations, a class of hydrophilic lipid derivatives—lysolipids—was incorporated to disrupt membrane integrity under heat. Lysolipid-based TSL (LTSL) demonstrated complete drug release within 2 s at 42 °C while maintaining sufficient stability in circulation. LTSL loaded with doxorubicin remains the only TSL formulation tested in human clinical trials. However, even with advancements, TSLs still exhibit a relatively short plasma half-life (approx. 1 h in human patients) due to residual leakage at body temperature. Recent studies indicate that incorporating 1,2-dipalmitoyl-sn-glycero-3-phosphoglyceroglycerol (DPPG2) improves plasma stability while maintaining rapid heat-triggered release [20]. Initial experiments using fluorescent dye encapsulation demonstrated an extended plasma half-life of 5–10 h in small animals, whereas doxorubicin-loaded DPPG2-TSL showed an in vivo plasma half-life of approximately 40 min in felines, comparable to the LTSL formulation.

## 5. Improving the Potential of Thermosensitive Liposomes—Thermosensitive Magnetoliposomes

Magnetic nanoparticles, particularly iron oxides such as Fe_3_O_4_ and γ-Fe_2_O_3_, have garnered significant attention in biomedical research due to their low toxicity, high biocompatibility, and magnetic properties. The most established clinical applications for iron oxides are contrast agents in magnetic resonance imaging (MRI). Moreover, they can be used as local heat generators in magnetic hyperthermia, offering a platform for controlled drug release [18]. The composition of the lipid bilayer in liposomal systems plays a crucial role in determining the phase transition temperature, which governs the temperature threshold for drug release (Figure 7). For instance, incorporating DSPC (Tm approx. 55 °C) into a DPPC-based formulation (Tm approx. 42 °C) effectively elevates the activation threshold for encapsulated drug release from approximately 35 °C to 40 °C. Thermosensitive magnetoliposomes (TSML), co-encapsulating iron oxide nanoparticles and cytotoxic agents such as arsenic trioxide (As_2_O_3_), have demonstrated near-complete drug release (approx. 98%) upon reaching a local temperature of 44 °C [18,19,20,21]. This heat-activated mechanism ensures controlled, site-specific drug liberation, minimizing systemic toxicity. TSML incorporating Fe_3_O_4_ nanoparticles can be externally activated by either alternating magnetic fields or near-infrared (NIR) laser irradiation (700–1000 nm). NIR light induces a localized photothermal effect, heating the liposomal environment to the transition point of the bilayer, promoting rapid drug release. Fe_3_O_4_ nanoparticles, known for their photothermal conversion efficiency, offer a straightforward and minimally invasive approach to heat generation when irradiated with NIR light, bypassing the need for more complex or potentially toxic photothermal agents such as gold nanorods or indocyanine green (ICG).

Gold-based nanostructures remain among the most widely used photothermal triggers for NIR-mediated drug release. TSL loaded with doxorubicin (DOX), physically co-formulated with gold nanorods, enhances therapeutic outcomes under NIR stimulation. However, Fe_3_O_4_-based systems are increasingly favored for their comparable photothermal efficiency, superior safety profile, and dual functionality, enabling both triggered release and MRI visualization.

Hydrophobic Fe_3_O_4_ nanoparticles are typically synthesized via thermal decomposition of Fe(acac)_3_—tris(acetylacetonato) iron (III)—in the presence of oleic acid and oleylamine in benzyl ether under an inert nitrogen atmosphere. Following purification and redispersion in hexane, the nanoparticles are incorporated into liposomes using the reverse-phase evaporation method [19,22,23,24,25]. The lipid matrix—commonly DPPC and cholesterol in a 4:1 molar ratio—is dissolved in chloroform and emulsified with ammonium sulfate buffer and Fe_3_O_4_ nanoparticles using probe sonication. After solvent removal and further sonication, uniform magnetoliposomes are obtained. DOX is actively loaded via an ammonium sulfate gradient (DOX:lipid ratio of 1:20) and incubated at 50 °C to facilitate drug encapsulation. Unencapsulated DOX is removed, and the final Fe_3_O_4_-TSL formulation is stored at 4 °C for further use. Biodistribution studies revealed that free ICG rapidly accumulates in tumors within 8 h post-injection but is quickly cleared thereafter. In contrast, ICG-loaded Fe_3_O_4_–TSL exhibits prolonged retention and progressive accumulation in tumor tissue, with a peak intratumoral concentration observed at 48 h post-administration. Ex vivo fluorescence imaging confirmed significantly stronger signal intensity in tumors treated with ICG-Fe_3_O_4_-TSL compared to those receiving free ICG. Quantitative analysis indicated that the fluorescence intensity in tumors was approximately twice as high in the liposomal group. These findings support the use of Fe_3_O_4_–TSLs as an effective drug delivery platform, with the optimal therapeutic window for NIR-triggered photothermal therapy occurring between 24 and 48 h post-injection [26,27].

## 6. Overcoming Incomplete Release Profiles—Enzyme-Responsive Liposomes

Liposomes distinguish themselves among nanocarriers due to their flexible drug-loading capabilities and ability to encapsulate a wide range of therapeutic agents. However, traditional liposomal systems often suffer from slow or incomplete release profiles, limiting their therapeutic utility. Enzyme-responsive liposomes address this disadvantage by allowing for precise, disease site-specific drug release driven by endogenous biological signals, eliminating the need for external activation methods [28,29]. This approach opens up new possibilities in oncology and other therapeutic areas, offering enhanced selectivity, minimized systemic toxicity, and improved treatment outcomes through more efficient site-specific drug delivery. Enzyme-responsive liposomes incorporate cleavable chemical motifs within the lipid bilayer or surface coatings, which are specifically recognized and processed by enzymes overexpressed at pathologically changed tissues [30]. Enzymatic cleavage initiates drug release through several mechanisms: structural degradation of the liposomal bilayer, removal of stabilizing PEG shells, or activation of drugs embedded within or attached to the liposome. Enzyme-responsive liposomes have been engineered to exploit the overexpression of specific hydrolases in the tumor microenvironment. Upon exposure to target enzymes such as esterases, β-galactosidases, or alkaline phosphatases, cleavable lipid conjugates undergo structural decomposition, leading to destabilization of the lipid bilayer and subsequent drug release. In vitro and in vivo studies report drug release efficiencies ranging from 80% to 95%, contingent on enzyme concentration, substrate specificity, and liposomal design. One notable system utilizes β-galactosidase-mediated cleavage of ganglioside GM1 analogs, where the removal of sugar moieties alters bilayer integrity. Similarly, the hydrolysis of phosphate-containing headgroups by tumor-associated alkaline phosphatase can trigger membrane disassembly. These mechanisms provide a high degree of site-specificity and are under active investigation in preclinical oncology pipelines. This approach enables the tailored delivery of therapeutics to sites rich in enzymes such as esterases, phosphatases, and β-galactosidases, which are frequently upregulated in various malignant tissues. Esterase activity in malignant tissues has been observed to be magnified by two to three orders of magnitude. Similarly, increased expression of alkaline phosphatase and glycosidases like β-galactosidase is commonly associated with tumorigenesis. A notable example involves β-galactosidase-mediated cleavage of the ganglioside GM1, where removal of the sugar moiety alters the membrane characteristics to prompt drug release. The hydrolysis of phosphate head groups by alkaline phosphatase also results in bilayer disruption, facilitating payload discharge. To refine enzyme-sensitive delivery strategies, Best and Lou [28] introduced a rational design of stimuli-responsive lipids tailored to specific enzymatic triggers through modular chemical engineering. Each lipid derivative consists of three functional segments (Figure 8):Enzyme-Specific Head Group: This segment is tailored to be cleaved by a particular enzyme—an ester group for esterases, a phosphate group for phosphatases, or a β-galactose for β-galactosidases [31,32,33].Self—Immolating Linker (SIL): Positioned between the lipid segment and the enzyme-sensitive group, the SIL undergoes rapid decomposition upon enzymatic cleavage, effectively destabilizing the membrane, offering rapid kinetic profiles.Membrane—Disruptive Lipid Backbone: The core lipid component, such as dioleoylphosphatidylethanolamine (DOPE) or aminodialkylglycerol, promotes membrane destabilization upon trigger-induced conversion into a non-bilayer-forming species. These lipids are selected based on both efficacy and synthetic tractability.

To assess whether the enzyme-responsive lipid could be selectively hydrolyzed, the compound can undergo a TLC assay. Additionally, enzyme activity for phosphatases and β-galactosidases can be verified through colorimetric assays employing *p*-nitrophenyl phosphate and *p*-nitrophenyl β-D-galactopyranoside, respectively, confirming the specificity and catalytic function of each enzymatic target [34].

## 7. Future Perspectives in Liposomal Drug Delivery

Future perspectives in liposomal drug delivery are focused on enhancing precision, overcoming immunological barriers, and integrating multifunctionality within a single carrier system. As research in nanomedicine continues to evolve, several key trends are emerging that will shape the next generation of liposome-based therapies. One such trend is the development of multi-responsive liposomes that can be programmed to react to a combination of internal and external stimuli. These stimuli may include pH changes, temperature shifts, enzyme activity, redox gradients, and even magnetic or ultrasound triggers. By incorporating multiple stimuli-responsive elements within a single vesicle, it becomes possible to achieve spatially and temporally controlled drug release, tailored specifically to the pathological environment of a given disease [35].

Biomimetic liposomes, cloaked with natural cellular membranes, represent a particularly promising platform for immune evasion and prolonged circulation. These “camouflaged” liposomes are typically coated with membranes derived from red blood cells, leukocytes, platelets, cancer cells, or exosomes. Such systems inherit membrane proteins and glycoproteins from their source cells, thereby offering homotypic targeting, reduced recognition by the mononuclear phagocyte system, and increased biocompatibility. For instance, erythrocyte membrane-coated liposomes have demonstrated significant improvements in circulation half-life, while cancer cell membrane-coated liposomes exhibit preferential accumulation in homologous tumors due to self-recognition mechanisms. A major advantage of these systems is their ability to navigate biological barriers by mimicking endogenous transport pathways. Nevertheless, challenges such as membrane integrity, scalability, and immune compatibility remain areas of active investigation [36].

Beyond biological mimicry, the integration of advanced computational tools such as artificial intelligence (AI) and machine learning (ML) is revolutionizing the rational design of liposomal formulations. Traditional trial-and-error methods are being replaced by predictive algorithms capable of modeling lipid–drug interactions, optimizing vesicle composition, and forecasting in vivo behavior. AI-driven approaches can analyze large datasets from preclinical and clinical studies to identify patterns that correlate physicochemical properties with therapeutic outcomes. Moreover, ML can support the selection of targeting ligands, linker chemistries, and surface modifications based on specific disease phenotypes. In silico formulation design not only accelerates development timelines but also reduces material consumption and cost, making personalized nanomedicine more attainable. Emerging platforms also integrate AI with microfluidic technologies, allowing real-time adjustment of formulation parameters during manufacturing.

Another frontier in liposomal innovation is the design of theranostic carriers—multifunctional vesicles that combine therapeutic payloads with diagnostic or imaging agents. Theranostic liposomes can be engineered to include fluorophores, MRI contrast agents, radionuclides, or photoacoustic markers, enabling real-time tracking of biodistribution and therapeutic response. This dual capability is particularly valuable in oncology, where it can inform adaptive treatment decisions and support the early detection of therapeutic resistance. For example, doxorubicin-loaded liposomes conjugated with near-infrared dyes have been used for simultaneous tumor ablation and intraoperative visualization. Similarly, liposomes labeled with PET isotopes can provide quantitative insights into pharmacokinetics and tumor uptake in vivo. Such systems not only enhance therapeutic efficacy but also offer critical safety advantages by ensuring that drugs primarily accumulate in the intended target tissue [36,37].

Regulatory and manufacturing considerations will play a decisive role in the clinical translation of future liposomal technologies. Despite their therapeutic potential, the complexity of multifunctional liposomes presents unique challenges in terms of quality control, batch reproducibility, and long-term stability. Regulatory agencies are increasingly encouraging the adoption of quality-by-design (QbD) frameworks, which emphasize risk-based development, critical quality attributes, and process analytical technologies (PAT). Advances in analytical techniques—such as cryo-electron microscopy (cryo-EM), nanoparticle tracking analysis (NTA), differential scanning calorimetry (DSC), and Raman spectroscopy—are providing deeper insights into vesicle structure, composition, and performance. In parallel, continuous manufacturing methods using microfluidic platforms are being explored as scalable alternatives to conventional batch processing. These innovations promise greater control over vesicle size, polydispersity, and encapsulation efficiency, thus improving the reliability and regulatory compliance of liposomal products.

Looking ahead, the convergence of nanotechnology, synthetic biology, computational science, and systems pharmacology will drive the next wave of breakthroughs in liposomal therapeutics. Researchers are already investigating programmable liposomes equipped with gene circuits, synthetic receptors, and logic-gated release systems. Such systems can “sense” their microenvironment and autonomously adjust their behavior in response to disease-specific cues. Additionally, the integration of omics data—such as genomics, proteomics, and metabolomics—into liposomal design may facilitate the development of truly personalized nanocarriers tailored to individual patients. While significant technical and regulatory hurdles remain, the future of liposomes is undeniably promising, with vast potential to transform drug delivery paradigms and redefine therapeutic precision [38].

## 8. Advanced Manufacturing Technologies and 3D-Printing Perspectives in Liposomal Drug Delivery

Despite recent advances, a major barrier in nanoliposome development remains the relatively low production efficiency of conventional microfluidic methods (typically a few mg/min). These methods utilize controlled fluid flow in microscopically sized channels, allowing for the precise formation of liposomes of defined size and composition [39,40,41]. Traditional laboratory-scale production methods often suffer from limited control over liposome properties (e.g., size and lamellar structure) and rely on laborious, multi-step procedures, which can limit preclinical development and innovation in this field. The widespread adoption of alternative, more controlled microfluidic methods is often hampered by the complexity and expense associated with manufacturing and operating the devices, as well as short device life and relatively low liposome production rates. By adjusting formulation and production parameters, including centrifugation time and speed, as well as lipid concentration, the average size of the produced liposomes could be tuned in the range of approx. 140 nm [42]. While the parallelization of manufacturing devices can enhance throughput, traditional manufacturing methods are often time-consuming, expensive, and reliant on specialist equipment such as syringe pumps and flow controllers. Moreover, microfluidic channels are prone to clogging and require tightly controlled environments, limiting their robustness in standard lab settings. The advent of three-dimensional (3D) printing as an additive liposome manufacturing technique has revolutionized the development of customizable drug delivery systems. To address manufacturing limitations, recent advances in 3D printing have facilitated the creation of cost-effective and rapid-to-produce microfluidic systems [41]. For instance, the reactor-in-a-centrifuge (RIAC) device developed by Cristaldi et al. represents a major step forward [39]. This system, manufactured within 8 h using a benchtop 3D printer and low-cost materials, is actuated via standard laboratory centrifuges (1000–2000 rcf), bypassing the need for complex flow control setups. With a single RIAC device achieving production rates of >8 mg/mL and scalable by parallel operation, this approach offers a compelling alternative for accessible liposome synthesis. Moreover, it achieves therapeutically relevant liposome sizes (<200 nm) with low polydispersity (PDI < 0.2). The use of 3D printing with digital light processing (DLP) enables the creation of complex and extremely accurate microfluidic systems, which enables the production of liposomal nanoparticles with consistent properties. This marks a notable advancement in lab-scale liposome manufacturing technologies. The integration of nanodrug delivery systems into 3D-printed matrices has been shown to influence critical factors such as matrix porosity, swelling behavior, and mechanical properties.

Liposomes can also be successfully incorporated into 3D-printed matrices composed of biofunctional polymers, such as chitosan, hydroxypropyl methylcellulose (HPMC), and polyvinyl alcohol (PVA). This integration enables sustained drug release from liposomal carriers, enhanced retention at the site of application (e.g., mucosal surfaces), and reduced systemic side effects due to localized action. Furthermore, the use of multilayered (core–shell) architectures allows for the modulation of drug release profiles and facilitates the simultaneous delivery of multiple therapeutic agents. Previous studies have demonstrated that the release kinetics of ibuprofen from polycarbonate–chitosan polymer blends can be tuned by adjusting the chitosan content [43]. Likewise, distinct doxycycline release behaviors were achieved using biphasic constructs—either drug-loaded cores or drug-loaded shells prepared from a single ink formulation. A promising example of such an approach is the incorporation of atorvastatin–propylene glycol liposomes (ATV/PG-Lip) into an innovative 3D-printed mucoadhesive buccal film (ATV/PG-Lip@3DP-film) [43]. This system was formulated using a blend of chitosan, PVA, and HPMC as the base matrix. To prepare the printing inks, PVA and HPMC were dissolved in phosphate buffer (pH 7.4), while chitosan was solubilized in 1% acetic acid. These polymer solutions were then blended with propylene glycol (20% *w/v*) to produce the base ink. Drug-loaded inks were obtained by incorporating either ATV in PG or preformulated ATV/PG-Lip into the polymer mixtures at optimized concentrations. The resulting composite 3D-printed films were characterized for their physicochemical properties and evaluated for antifungal activity both in vitro and in vivo using a rabbit model of oral candidiasis. The ATV/PG-Lip formulation exhibited a favorable nanoscale size distribution, efficient drug encapsulation, high colloidal stability over a three-month period, and in vitro cytocompatibility. Moreover, the ATV/PG-Lip@3DP-film demonstrated controlled water uptake, disintegration, and drug release, along with excellent mucoadhesive properties. These results highlight the potential of integrating nanodrug delivery systems and 3D-printing technologies to develop patient-friendly, self-administered therapeutic platforms capable of effectively addressing antifungal drug resistance [39,40,41].

## 9. Conclusions

In recent decades, nanocarriers have evolved from simple passive drug delivery systems to highly engineered, multifunctional molecules tailored for precision medicine. The unique structural composition of liposomes—biocompatible lipid bilayers capable of encapsulating both hydrophilic and hydrophobic drugs—combined with their tunable physicochemical properties, has enabled application in oncology, infectious diseases, and inflammation-driven pathologies. One of the foundational advantages of liposomes is their ability to passively accumulate in pathological tissues via the enhanced permeability and retention (EPR) effect. This passive targeting approach has demonstrated superior pharmacokinetics compared to free drugs, allowing for reduced off-target toxicity and improved therapeutic indices. However, the EPR effect alone has limitations due to heterogeneous vascular permeability among tumors and poor lymphatic drainage. Despite promising advances in modern liposome modifications, several challenges remain unsolved, and a clinical transition cannot be fully realized. Heterogeneity in tumor vasculature and enzyme expression may limit uniform drug release and must be studied further. Despite promising outcomes of scientific studies, long-term safety and potential immunogenicity must be taken into account, particularly for systems employing metallic nanoparticles or non-natural enzyme-responsive linkers.

Future directions in liposome research are likely to focus on several critical fronts. An interesting idea that attracts the attention of the scientific community is the potential capability of liposomes to respond to multiple stimuli at once for enhanced spatial and temporal control over drug release. The increased use of artificial intelligence yields a promising field for guided formulation design, optimizing lipid composition and targeting ligands based on predictive modeling and machine learning algorithms.

While clinical implementation still faces several hurdles, the integration of stimulus-responsive elements—particularly those leveraging magnetic and enzymatic triggers—marks a significant leap forward in developing next-generation therapeutics. A multidisciplinary approach encompassing materials science, pharmacology, molecular biology, and clinical sciences will be essential to unlock the full potential of liposomal technologies in personalized medicine. Future directions in liposomal drug delivery are moving toward multifunctional and highly personalized systems. A key trend involves the integration of multiple stimulus-responsive elements within a single carrier platform. Such systems are capable of responding sequentially or simultaneously to environmental cues such as pH, enzymatic activity, oxidative stress, or temperature gradients.

## Figures and Tables

**Figure 1 pharmaceutics-17-00885-f001:**
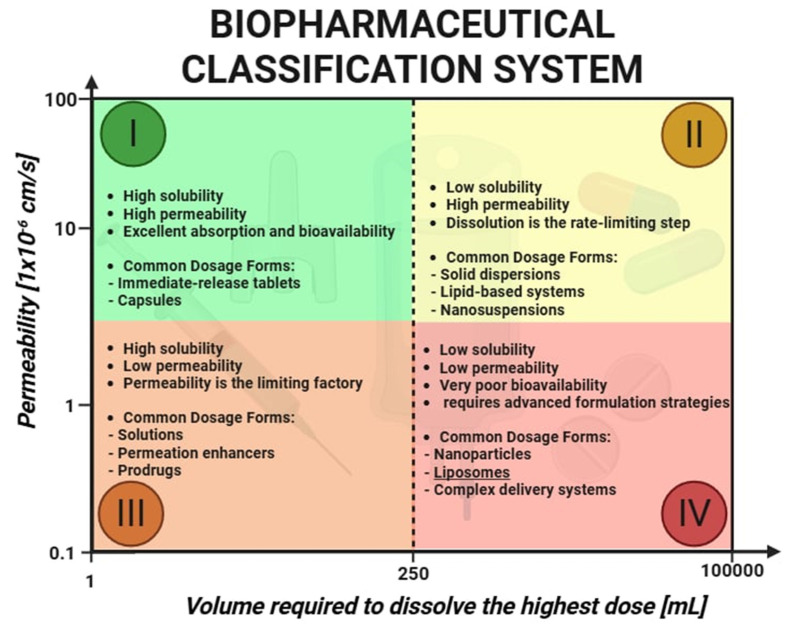
Visualization of Noyes–Whitney equation combined with Biopharmaceutics Classification System.

**Figure 2 pharmaceutics-17-00885-f002:**
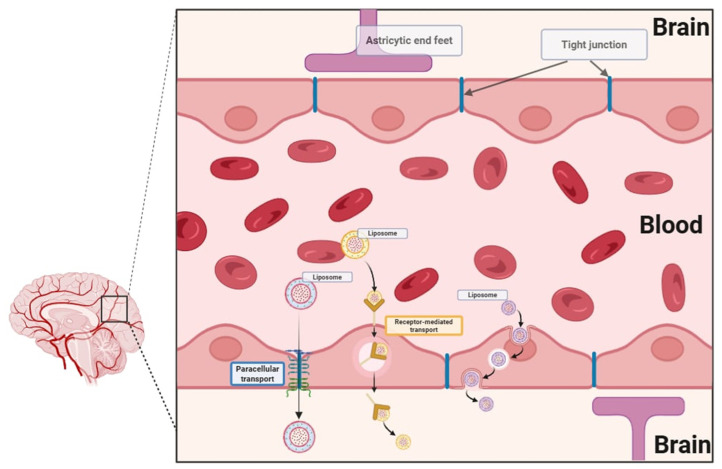
Transport of xenobiotics across the blood–brain barrier.

**Figure 3 pharmaceutics-17-00885-f003:**
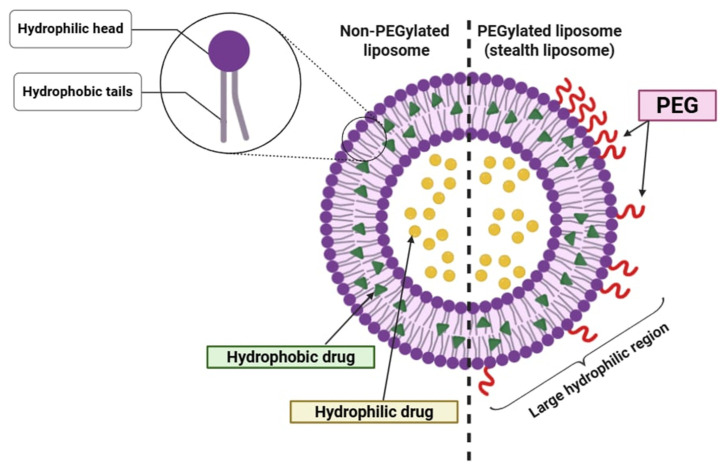
Stealth liposome structure scheme.

**Figure 4 pharmaceutics-17-00885-f004:**
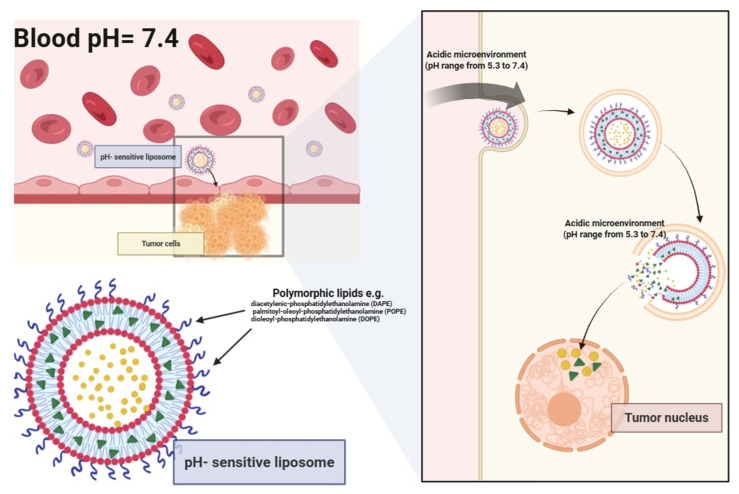
Mechanism of action of pH-sensitive liposomes.

**Figure 5 pharmaceutics-17-00885-f005:**
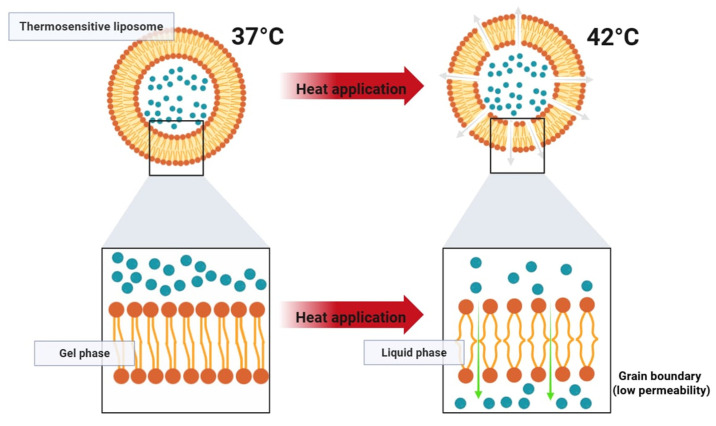
Mechanism of action of thermosensitive liposomes.

**Figure 6 pharmaceutics-17-00885-f006:**
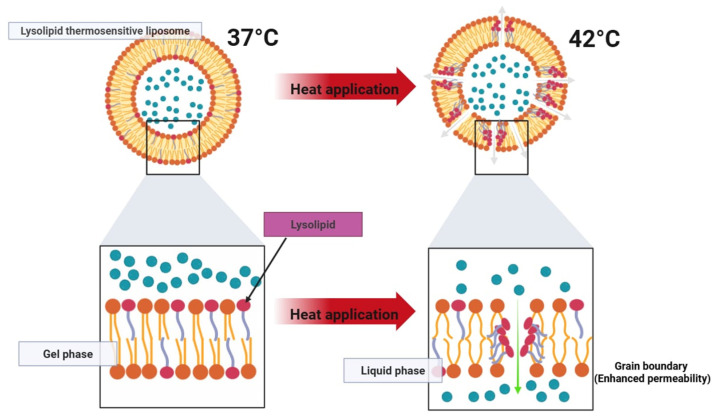
Mechanism of action of lysolipid thermosensitive liposomes.

**Figure 7 pharmaceutics-17-00885-f007:**
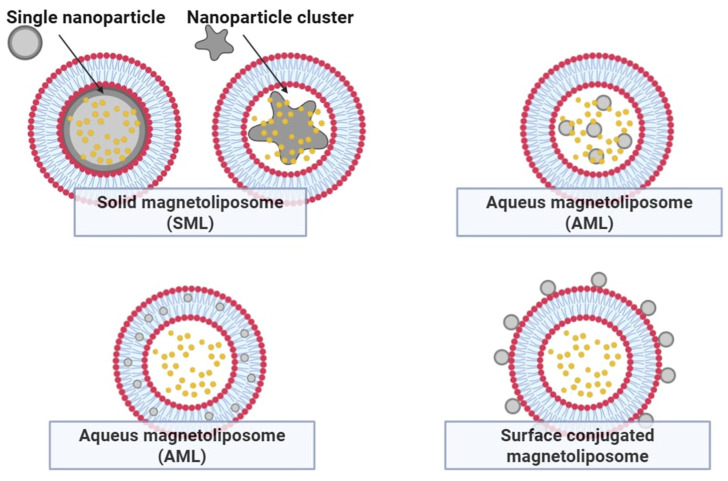
Types of magnetoliposome structures.

**Figure 8 pharmaceutics-17-00885-f008:**
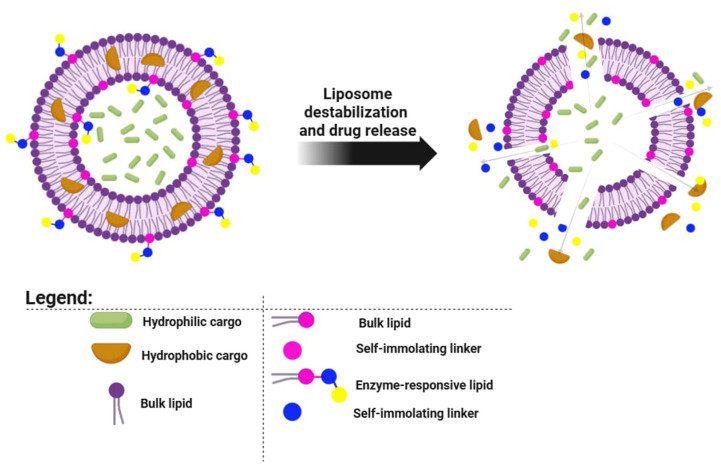
Design and mechanism of action of stimuli-responsive lipid.

## Data Availability

No new data were created.

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
