# Peer review of "Advances in Liposomal Drug Delivery: Multidirectional Perspectives on Overcoming Biological Barriers"

_pharmaceutics, 2025, doi:10.3390/pharmaceutics17070885_

Round 1

Reviewer 1 Report (Previous Reviewer 2)

Comments and Suggestions for Authors

This review entitled "Advances in Liposomal Drug Delivery: Multidirectional Perspectives on Overcoming Biological Barriers" summarized the research advances in classic and safe liposomal drug delivery. Stealth Liposomes,pH-Sensitive Liposomes,Thermosensitive Liposomes,Thermosensitive Magnetoliposomes,Enzyme - Responsive Liposomes are described in the review.

In this resubmitted manuscript, comparing to the previous submit, this one extended two parts," 7. Future perspectives in liposomal drug delivery" and " 8. Future perspectives in liposomal drug delivery". I don't know why these two titles are the same. Maybe it is a mistake.

Part 7 have no reference. Is this normal or there is no need to cite the reference?

The format of part 7,8, is different from other part of the manuscript.

The other parts of this review are totally well written. 

Author Response

Dear Reviewer,

Thank you very much for your thorough and insightful review of our manuscript. We greatly value your encouraging comments and are grateful for your recognition of the manuscript’s scientific contribution.We would like to clarify the concerns you raised:

Duplicate Titles (Sections 7 & 8):
Thank you for noticing the repetition in section titles. This was indeed a formatting oversight. The title of Section 8 has now been corrected to:
“Advanced Manufacturing Technologies and 3D-Printing Perspectives in Liposomal Drug Delivery”
This section focuses on the application of microfluidics and additive manufacturing, in contrast to Section 7 which covers future biological and AI-guided innovations.

Missing References in Section 7:
You rightly observed that Section 7 initially lacked citations. We have now added appropriate references to support each claim regarding biomimetic liposomes, AI integration, and theranostics. These additions strengthen the credibility of this section and ensure consistency across the manuscript.

Inconsistent Formatting in Sections 7 & 8:
We appreciate your feedback regarding the formatting differences. Both sections have now been revised to align stylistically and structurally with the rest of the manuscript, maintaining a uniform academic tone and consistent paragraph structure.

Once again, thank you for your valuable feedback. Your comments helped us significantly improve the manuscript, and we hope that the revised version meets the high standards expected by Pharmaceutics.

With sincere appreciation,
Rafał Chiczewski
On behalf of the authors
Department of Pharmaceutical Technology
Pomeranian Medical University in Szczecin

Reviewer 2 Report (Previous Reviewer 4)

Comments and Suggestions for Authors

The authors have successfully addressed the most concerns, responded to the comments and made the necessary corrections at the text. The manuscript has been sufficiently improved and now warrants publication in Pharmaceutics, therefore I recommend the publication of this work.

Author Response

Dear Reviewer,

On behalf of all co-authors, I would like to express our sincere gratitude for your time and constructive evaluation of our manuscript entitled "Advances in Liposomal Drug Delivery: Multidirectional Perspectives on Overcoming Biological Barriers."

We appreciate your positive recommendation and acknowledgment that the revised manuscript has successfully addressed the prior concerns and now warrants publication. Your feedback was very encouraging and has helped us further refine the quality of our work.

As indicated, we have carefully revised the text, improved the clarity of key sections, and ensured that all technical corrections and formatting updates were addressed. We are pleased to hear that the English language and manuscript structure were satisfactory.

Thank you again for your contribution to improving our manuscript.

Warm regards,
Rafał Chiczewski
On behalf of the authors
Department of Pharmaceutical Technology
Pomeranian Medical University in Szczecin

This manuscript is a resubmission of an earlier submission. The following is a list of the peer review reports and author responses from that submission.

Round 1

Reviewer 1 Report

Comments and Suggestions for Authors
  1. The review does not present any new insights, perspectives, or syntheses of the existing literature. It largely reiterates established facts about liposomal drug delivery systems without offering a critical analysis or unique viewpoint.
  2. The review may lack a clear and logical flow. The sections may not be well-organized, and the transitions between them may be abrupt or unclear, making it difficult for the reader to follow the main arguments.
  3. Similar reviews already available https://pubs.rsc.org/en/content/articlehtml/2021/xx/d0nh00605j, https://www.mdpi.com/2073-4360/14/5/925 
  4. Targeted Drug Delivery - pH-Sensitive Liposomes , only one citation for this entire paragraph Line no 133 to 172. 
  5. Line no 180-221 no citations found. 
  6. Poor referencing throughout the entire manuscript. 
  7. Challenges and future prospects is missing. Clinical trials and patent data is also missing from the manuscript.
  8. In depth applications are missing from the manuscript. 
Comments on the Quality of English Language

The hyphenation of terms like "nanocarrier" and "non - encapsulated" is inconsistent throughout the text.

Some sentences are lengthy and complex, which could be simplified for better readability. Additionally, there are a few instances where grammatical errors are present.

Author Response

Thank You for Your insightful and constructive comments. We sincerely appreciate Your thorough review and the valuable suggestions You have provided. Your feedback has helped us improve the clarity and quality of our work, and we have carefully addressed each of Your points in the revised version. We are grateful for Your time and effort in reviewing our manuscript.

  1. We would like to express our sincere gratitude for your thorough analysis and insightful comments. We truly appreciate the depth of your experience, which has significantly contributed to improving our manuscript. Thank you for dedicating considerable time to this review and for providing such a prompt response. We acknowledge that our review may not introduce groundbreaking findings to the scientific community, and similar publications likely exist. However, regular updates of review articles are essential, given the rapidly evolving nature of scientific reports in the field of drug formulation technology. Creating a comprehensive review is a highly time-intensive process, requiring deep analysis and specialized knowledge, supported by extensive professional experience.

At this stage, we are unable to offer specific clinical recommendations, as our work primarily focuses on basic scientific research rather than clinical applications. Nonetheless, we hope that the findings presented in this review will serve as a valuable foundation for future studies, potentially paving the way for clinical translation. Our aspiration is to contribute to the transformation of scientific advancements into tangible improvements in diagnostic standards and less invasive therapeutic options for patients.

  1. We agree that the overall scope may appear unclear. To make it easier for readers to understand the essence of the work, we have organized the subchapters and narrowed their topics to focus only on the most important aspects of liposomal drug delivery systems (lines 29-55, 143-166, 205-213, 257-267, 351-368, 397-470). The comment on the critical analysis did a great influence improving our work. We agree that the earlier version focused more on a reproducible summary of the work and lacked the authors' deeper analysis. The entire section “Future perspectives in liposomal drug deliveryhas been added and focuses on a broader analysis of the challenges posed by liposomal drug delivery systems and will allow readers to acquire their own reflections (lines 397-470).
  2. Agreed and answered on point 1 and 2.

4/5/6. We also thoroughly analyzed the comments regarding the bibliography - we expanded the scope of citations and filled in the gaps indicated in the comments (lines 180 – 230). We also have taken the opportunity to MDPI Author Services language editing in an effort to improve the standard of English, readability and clarity throughout. We feel that this has improved the standard of this submission significantly.

7/8.  Unfortunately, at this moment we cannot provide any further clinical recommendations, because our work is mainly focused on research for scientific purposes, not in clinical practice. However, we hope that the studies conducted for the review article will soon be able to be used as a starting point for research work. We very much hope that we will be able to be part of the translation of scientific achievements into real changes in clinical practice - and consequently - improvement of diagnostic standards and less invasive therapies for patients. We agree that the earlier version focused on a reproducible summary of the work and lacked the authors' deeper analysis. The entire section “Future perspectives in liposomal drug deliveryhas been added and focuses on a broader analysis of the challenges posed by liposomal drug delivery systems and will allow readers to acquire their own reflections (lines 397-470).

Reviewer 2 Report

Comments and Suggestions for Authors

In this review, multidirectional perspectives of liposomal drug delivery on overcoming biological barriers was addressed. In the abstract part, it seems that the biological barriers were not described, and how the liposomal drug overcomed them also need to be addressed.

Usually, the biological barriers is the main problem the liposomes will meet. This angel they choose is good for the review. liposomes is a very classic and safe nano materials. Many liposomes drugs are now on the market or clinic trails. This review updated the newly published research progress. The expert in this field would harvest a lot when the read this review.

The biological barriers include many other situations the liposomes will meet. Or the authors in this review should change the title. The main subject in this review is the shotcomings of liposomes and how they modifed liposomes to achieve certain goals, not just to overcome the biological barriers. How liposomes overcome their own shortcomings in Liposomal Drug Delivery seems to be the main point.

In this reviewm, the references cited is OK.

For the figures, in figure 5, 6, why not combine them together?

Author Response

Thank You for Your insightful and constructive comments. We sincerely appreciate Your thorough review and the valuable suggestions You have provided. Your feedback has helped us improve the clarity and quality of our work, and we have carefully addressed each of Your points in the revised version. We are grateful for Your time and effort in reviewing our manuscript.

  1. We agree that presenting a too general overview of biological barriers may appear unclear. To address this, our numerous figures were specifically designed to illustrate the challenges of xenobiotic penetration through these barriers. Additionally, to further enhance the clarity and impact of our work, we have introduced several changes and included specific examples of biological barriers to provide readers with a more comprehensive understanding. (lines 29 – 55).

2/3/4. Thank you for your insightful comment regarding the proposed title change. We acknowledge that our work addresses a broader spectrum of contemporary liposome modifications. However, we believe that the core focus of these modifications is ultimately to overcome biological barriers, which inherently addresses the limitations of the nanocarriers themselves. While these topics are closely related, based on our prior experience with scientific publications, we prefer not to broaden the title’s scope, as this might lead to potential misunderstandings about the specific focus and detailed content of the article.

  1. Both figures indeed cover a similar scope, and we appreciate your observation. However, we have deliberately chosen to present them separately to avoid potential text illegibility. We believe that this approach is technically preferable, as it ensures clarity and allows readers to fully appreciate the quality of the figures we have prepared.

Reviewer 3 Report

Comments and Suggestions for Authors

This review provides a broad overview of the evolution and current state of liposomal drug delivery systems, focusing on various strategies developed to overcome biological barriers and achieve targeted therapy. It covers fundamental concepts like the EPR effect and passive targeting, progresses through established modifications like PEGylation (stealth liposomes), and discusses several types of stimuli-responsive systems (pH, temperature, enzyme, magnetic). The scope is comprehensive, and the manuscript serves as a good introductory or summary text for those seeking a wide-angle view of liposomal technologies. However, some revision is warranted:

  1. In the introduction, when discussing the benefits of liposomes, try to focus on liposomes only instead of mentioning nanocarriers in general. Try to use more liposome-specific references as well.
  2. Be careful and critical about mentioning the "EPR" effect, as many clinical studies have shown questions in humans. Please add a more critical introduction/discussion regarding the EPR effect by citing the relevant literature
  3. There are lots of reviews in the literature covering the topics of liposomal drug delivery systems. In the introduction, please state the scope, novelty, and purpose of this review. If it is an updated review, please provide more recent references and elaborate on the discussion of the targeted liposomal system, the latest research on more key topics that are missing in the manuscript, such as ligand conjugation, prodrug strategies, etc.
  4. Perform a final proofread for minor grammatical errors or awkward phrasing.
  5. Check if Figures 5 and 6 can be presented more concisely or if their distinct points need slightly clearer emphasis.
  6. The review tends to be more descriptive than critical. While advantages are highlighted, a more balanced discussion incorporating the specific drawbacks, complexities in large-scale manufacturing, regulatory challenges, and reasons for the limited clinical translation of many advanced stimuli-responsive systems would strengthen the manuscript significantly.

Author Response

Thank You for Your insightful and constructive comments. We sincerely appreciate Your thorough review and the valuable suggestions You have provided. Your feedback has helped us improve the clarity and quality of our work, and we have carefully addressed each of Your points in the revised version. We are grateful for Your time and effort in reviewing our manuscript.

  1. We sincerely appreciate your valuable feedback regarding the use of the term "nanocarriers" and the scope of the bibliography. In response, we have refined the bibliography to focus specifically on liposomal modifications, significantly narrowing its scope from the broader concept of nanocarriers. We believe this adjustment will enhance the clarity of our work and make it easier for readers to locate relevant references. Thank you for this insightful suggestion, which has positively impacted the overall readability and coherence of our manuscript.

(lines 29-55)

  1. We also thoroughly analyzed the comments regarding the bibliography - we expanded the scope of citations and filled in the gaps indicated in the comments (lines 180 – 230). We also have taken the opportunity to MDPI Author Services language editing in an effort to improve the standard of English, readability and clarity throughout. We feel that this has improved the standard of this submission significantly.
  2. Both figures indeed cover a similar scope, and we appreciate your observation. However, we have deliberately chosen to present them separately to avoid potential text illegibility. We believe that this approach is technically preferable, as it ensures clarity and allows readers to fully appreciate the quality of the figures we have prepared.
  3. We hope that the studies conducted for the review article will soon be able to be used as a starting point for research work. We very much hope that we will be able to be part of the translation of scientific achievements into real changes in clinical practice - and consequently - improvement of diagnostic standards and less invasive therapies for patients. We agree that the earlier version focused on a reproducible summary of the work and lacked the authors' deeper analysis. The entire section “Future perspectives in liposomal drug deliveryhas been added and focuses on a broader analysis of the challenges posed by liposomal drug delivery systems and will allow readers to acquire their own reflections (lines 397-470).

Reviewer 4 Report

Comments and Suggestions for Authors

The submitted review makes a good impression. The authors clearly describe the main modern achievements in the application of liposomes, including stimulu-sensitive, for drug delivery without overloading with unnecessary information. I recommend review for publication after addressing the following.

  1. In my opinion, it would be useful to add detailed information on the disadvantages of liposomes as drug delivery systems.
  2. Section 2 should include information on new trends in replacing PEG with other polymers. According to new studies, PEG is not inert to living organisms.
  3. It would be helpful if authors provide information on release rates under pH changing. What is the efficacy? Is all encapsulated substance released? The same information seems useful when describing the effects of other stimuli

Author Response

Thank You for Your insightful and constructive comments. We sincerely appreciate Your thorough review and the valuable suggestions You have provided. Your feedback has helped us improve the clarity and quality of our work, and we have carefully addressed each of Your points in the revised version. We are grateful for Your time and effort in reviewing our manuscript.

1 / 2.  We would like to express our sincere gratitude for your thorough analysis and insightful comments. We truly appreciate the depth of your experience, which has significantly contributed to improving our manuscript. Thank you for dedicating considerable time to this review and for providing such a prompt response. We would like to express our sincere gratitude for your thorough analysis and insightful comments. This is a very valid point regarding PEGylation. We have taken this valuable feedback into account and included the necessary clarification. (lines 149 – 172)

  1. We have taken your precious comment into account and included missing information. (lines 211 – 219, 257 – 267, 363 – 374)

Reviewer 5 Report

Comments and Suggestions for Authors

It seems to me that the manuscript provides an overview of the development of liposomes, but it lacks significant organization and primarily offers relatively basic introductory content. Its academic value is limited.

1.The Introduction part lacks logical coherence. First, the author discusses the poor solubility of most drugs, then mentions the EPR effect in tumors and characteristics of inflamed sites, followed by a discussion of the blood-brain barrier, and finally introduces liposomes. However, when addressing liposomes, the authors only briefly mention the advantage of their particle size. The Introduction fails to provide readers with a coherent background on liposomes.

2.When discussing the different functions of liposomes, the authors do not comprehensively include existing understandings or the latest advancements in the field. For example, regarding stealth liposomes modified with PEG. while the article highlights the advantages of PEGylation (e.g., prolonged circulation), it neglects to mention the limitations of PEGylated liposomes, such as steric hindrance-related issues or the potential for the accelerated blood clearance phenomenon induced by PEG, and how to solve the problems. Additionally, when mentioning PEG conjugation to proteins or peptides for targeted delivery, the authors should subcategorize or elaborate on the diverse types of peptides/proteins used in such modifications to deepen the analysis.

3.The authors do not thoroughly explore individual topics or structure the content systematically. Some functionalized stratedies could be discussed in the review. For instance, biomimetic membranes, which mimic liposomal properties but offer additional advantages, represent an important recent advancement in the field. These innovations and their implications for future development should be added as Future perspective part, which are entirely absent from the discussion, limiting the article’s comprehensiveness and relevance to current research trends.

Author Response

We would like to express our sincere gratitude for Your thoughtful and detailed review. Your insightful comments and suggestions have been invaluable in enhancing the quality and clarity of our work. The constructive feedback has not only strengthened this manuscript but has also been a valuable learning experience for us, which will undoubtedly improve our future research and publications. We truly appreciate the time and effort You have dedicated to this review and are deeply grateful for Your contributions.

  1. We would like to express our sincere gratitude for your thorough analysis and insightful comments. We truly appreciate the depth of your experience, which has significantly contributed to improving our manuscript. Thank you for dedicating considerable time to this review and for providing such a prompt response. We agree that presenting a too general overview of biological barriers may appear unclear. To address this, our numerous figures were specifically designed to illustrate the challenges of xenobiotic penetration through these barriers. Additionally, to further enhance the clarity and impact of our work, we have introduced several changes and included specific examples of biological barriers to provide readers with a more comprehensive understanding. (lines 29 – 55). We agree that the overall scope may appear unclear. To make it easier for readers to understand the essence of the work, we have organized the subchapters and narrowed their topics to focus only on the most important aspects of liposomal drug delivery systems (lines 29-55, 143-166, 205-213,257-267, 351-368, 397-470). We acknowledge that our work addresses a broader spectrum of contemporary liposome modifications. However, we believe that the core focus of these modifications is ultimately to overcome biological barriers, which inherently addresses the limitations of the nanocarriers themselves. The entire section “Future perspectives in liposomal drug deliveryhas been added and focuses on a broader analysis of the challenges posed by liposomal drug delivery systems and will allow readers to acquire their own reflections (lines 397-470).
  2. We would like to express our sincere gratitude for your thorough analysis and insightful comments. This is a very valid point regarding PEGylation. We have taken this valuable feedback into account and included the necessary clarification. (lines 149 – 172). We have taken your precious comment into account and included missing information. (lines 211 – 219, 257 – 267, 363 – 374)

To make it easier for readers to understand the essence of the work, we have organized the subchapters and narrowed their topics to focus only on the most important aspects of liposomal drug delivery systems (lines 29-55, 143-166, 205-213, 257-267, 351-368, 397-470). We acknowledge that our work addresses a broader spectrum of contemporary liposome modifications. However, we believe that the core focus of these modifications is ultimately to overcome biological barriers, which inherently addresses the limitations of the nanocarriers themselves. The entire section “Future perspectives in liposomal drug deliveryhas been added and focuses on a broader analysis of the challenges posed by liposomal drug delivery systems and will allow readers to acquire their own reflections (lines 397-470). We acknowledge that our review may not introduce groundbreaking findings to the scientific community, and similar publications likely exist. However, regular updates of review articles are essential, given the rapidly evolving nature of scientific reports in the field of drug formulation technology. Creating a comprehensive review is a highly time-intensive process, requiring deep analysis and specialized knowledge, supported by extensive professional experience. At this stage, we are unable to offer specific clinical recommendations, as our work primarily focuses on basic scientific research rather than clinical applications. Nonetheless, we hope that the findings presented in this review will serve as a valuable foundation for future studies, potentially paving the way for clinical translation.

Round 2

Reviewer 1 Report

Comments and Suggestions for Authors

The author has addressed all the reviewer’s comments. The manuscript is accepted for the publication. 

Reviewer 3 Report

Comments and Suggestions for Authors

The authors have addressed my comments accordingly and I would recommend acceptance of the manuscript.

Reviewer 5 Report

Comments and Suggestions for Authors

The manuscript offers a thorough overview of liposome research, serving as a valuable resource for novice researchers. However, it places excessive emphasis on foundational concepts, limiting its utility for readers seeking insights into cutting-edge advancements. While bionic technologies are mentioned in this version, other  key frontier areas, such as multifunctional integrated systems, 3D printing technology, are not mentioned.

Second, the "Future perspectives in liposomal drug delivery" section should be structured with clear subheadings to systematically address technological frontiers.